# Interconnections between the Cation/Alkaline pH-Responsive Slt and the Ambient pH Response of PacC/Pal Pathways in *Aspergillus nidulans*

**DOI:** 10.3390/cells13070651

**Published:** 2024-04-08

**Authors:** Irene Picazo, Eduardo A. Espeso

**Affiliations:** Department of Molecular and Cellular Biology, Centro de Investigaciones Biológicas (CIB) Margarita Salas, Spanish Research Council (CSIC), Ramiro de Maeztu, 9, 28040 Madrid, Spain; irene.picazo@cib.csic.es

**Keywords:** signalling, post-translational modifications, transcriptional factors, cross regulation, abiotic stress

## Abstract

In the filamentous ascomycete *Aspergillus nidulans*, at least three high hierarchy transcription factors are required for growth at extracellular alkaline pH: SltA, PacC and CrzA. Transcriptomic profiles depending on alkaline pH and SltA function showed that *pacC* expression might be under SltA regulation. Additional transcriptional studies of PacC and the only pH-regulated *pal* gene, *palF*, confirmed both the strong dependence on ambient pH and the function of SltA. The regulation of *pacC* expression is dependent on the activity of the zinc binuclear (C6) cluster transcription factor PacX. However, we found that the ablation of *sltA* in the *pacX^−^* mutant background specifically prevents the increase in *pacC* expression levels without affecting PacC protein levels, showing a novel specific function of the PacX factor. The loss of *sltA* function causes the anomalous proteolytic processing of PacC and a reduction in the post-translational modifications of PalF. At alkaline pH, in a null *sltA* background, PacC^72kDa^ accumulates, detection of the intermediate PacC^53kDa^ form is extremely low and the final processed form of 27 kDa shows altered electrophoretic mobility. Constitutive ubiquitination of PalF or the presence of alkalinity-mimicking mutations in *pacC*, such as *pacC^c^14* and *pacC^c^700*, resembling PacC^53kDa^ and PacC^27kDa^, respectively, allowed the normal processing of PacC but did not rescue the alkaline pH-sensitive phenotype caused by the null *sltA* allele. Overall, data show that Slt and PacC/Pal pathways are interconnected, but the transcription factor SltA is on a higher hierarchical level than PacC on regulating the tolerance to the ambient alkalinity in *A. nidulans*.

## 1. Introduction

Among the eukaryotic organisms, the fungal kingdom stands out for its wide species diversity, colonising diverse niches that often require an adaptive capacity due to adverse variations in environmental conditions. In filamentous fungi, such as the model organism *Aspergillus nidulans*, several transcriptional factors are involved in the regulation of gene expression in response to environmental pH and/or elevated concentrations of ions [1]. How these organisms manage stress caused by changes in ambient conditions is of general interest in biotechnology, agriculture, breeding and clinics, since pathogens use this capacity to favour infections in plants and animals [2,3,4].

Abiotic stresses such as temperature, high cation concentration, pH, UV radiation or reactive oxygen species represent a challenge for cell survival and molecular machinery responds by detoxifying the cell, repairing damaged DNA and maintaining cellular homeostasis [2,5,6,7,8]. In *A. nidulans*, three Cys2-His2 (C2H2) zinc-finger transcription factors play a major role in response to ambient pH alkalinisation: SltA, PacC and CrzA [1]. This class of transcription factors (TF) is widely found from yeast to mammals presenting a highly conserved DNA-binding motif, which for each TF recognises specific sequences in a promoter to enhance or inhibit gene expression [9,10]. The best studied regulator of the response to alkalinity in Ascomycota phylum is the 674-amino-acid transcription factor PacC [11,12,13,14], also named RIM101 in yeast-like fungi [15,16,17]. In *A. nidulans*, PacC harbours a three finger DNA-binding domain between residues 76 and 175 [9,18]. At an acidic pH, the primary form PacC^72kDa^ is mainly detected, but at an alkaline pH, the Pal pathway is activated and drives the proteolytic processing of this transcription factor [19,20]. When ambient pH alkalinises, the transmembrane sensor protein PalH, assisted by PalI [4], induces the phosphorylation and ubiquitination of the arrestin-like PalF [21]. These post-translational modifications (PTMs) in PalF promote the recruitment of the ESCRT-I Vps23 protein through PalF SxP motifs [22,23]. Vps23 connects Pal pathway proteins with other elements of ESCRT complexes, leading to the recruitment of PalA, which directly interacts with PacC^72kDa^ through the two YPxL/I motifs present in PacC [24]. The PalB protease binds to this complex and truncates the PacC^72kDa^ form between residues 493 and 500, rendering the 53 kDa intermediate form [25,26]. The PacC^53kDa^ form is an intermediary committed to a second, pH-independent, processing step by the proteasome, yielding the active PacC^27kDa^ form, consisting of approximately 254 residues [1,25,27]. PacC^27kDa^ has a dual regulatory role, enhancing the expression of genes while repressing others at alkaline pH [28,29]. Some of the genes regulated by this transcriptionally active form encode for extracellular enzymes (acid and alkaline phosphatases or proteases), permeases (*gabA*), transporters (*enaA* and *enaB*) and secondary metabolites (penicillin biosynthetic cluster), in addition to the *pacC* gene itself [4,30]. Limited knowledge on the regulation of *pacC* expression is available. In some Pezizomycotina, a negative regulator of *pacC* expression was described. PacX is a 661-residue protein, only found in *Leotiomycetes*, which has an atypical structure within the fungal Zn(II)2Cys6 transcription factor family, with its DBD located in the C-terminal region between residues 445 and 472 and a coiled-coil structure close to the N-terminal region [31,32,33]. The absence of PacX activity deregulates *pacC* expression by increasing its transcription and protein levels [31]. However, the activation/signalling pathway and other cellular functions of PacX remain unknown.

The 698-amino-acid SltA protein is another one of the C2H2 zinc-finger transcription factors involved in the response to environmental alkalinity, calcium stress and to a high salt concentration [34,35,36,37,38] and in determining *Aspergillus fumigatus* virulence, germination and hyphal development [39,40]; and, it is identified as ACE1 in *Trichoderma reesei*, regulating the expression of cellulases and xylanases [41,42]. The three zinc-finger DNA-binding domain is located between residues 416 and 500 [38]. As also seen with PacC, SltA displays three forms in the cell, the primary form of 78 kDa, SltA^78kDa^, and two proteolysed forms of 32 kDa, one of which is phosphorylated (SltA^32kDa^ + P and SltA^32kDa^, [43]). Currently, the Slt regulatory system comprises the transcriptional factor SltA and the signalling protease SltB, both being exclusive of Pezizomycotina subphylum [43]. SltB is a 1272-amino-acid protein consisting of a pseudo-kinase domain, possibly acting as a prodomain, and a serine-protease domain [43]. The protease activity of SltB allows its auto-proteolysis by releasing the pseudo-kinase prodomain and, subsequently, the proteolytic processing of SltA^78kDa^ to SltA^32kDa^. This processed 32 kDa form of SltA is necessary for the expression of *sltB* and *sltA* itself [38,43].

Initial transcriptional analyses using Northern blotting have evidenced the dual role for SltA^32kDa^, acting positively or negatively over a variety of genes encoding cation homeostasis transporters, such as *enaA*, *vcxA*, *pmcA* and *pmcB*, or regulatory elements involved in morphogenesis and secondary metabolism pathways [1,38,44]. An RNA-sequencing study increased our knowledge on these dual activities [44]. Transcriptomic profiles when comparing growth under high sodium concentration and alkaline pH conditions showed the strong differences between the wild-type and null *sltA* strains. Differential gene expression was observed in genes encoding mainly for transporter proteins of the Major Facilitator Superfamily (MFS), NAD (P)-binding domain-containing proteins and Zn(II)2Cys6 transcription factors [45], which were usually different depending on the type of stress induced. This response was highly altered in the absence of SltA, affecting clusters of secondary metabolism pathways such as the penicillin or sterigmatocystin biosynthesis pathway under stress conditions. Interestingly, among the genes that lost differential expression in a null *sltA* background was *pacC* [44].

In this work, we analysed the hierarchical relationship between Slt and Pal/PacC regulatory systems at alkaline pH by combining classical *pal/pacC* and *pacX* mutants with those of the Slt pathway. We observed an epistatic effect of SltA on the PacC/PacX/Pal system that goes beyond the transcriptional level. Severe deficiencies in SltA activity led to anomalous processing of PacC and restoring constitutive ambient alkaline pH signalling, or the presence of a constitutive PacC active form in the fungal cell was not sufficient to recover wild-type tolerance to alkaline pH in the absence of SltA. Thus, this study presents an additional control in Pezizomycotina to the Pal/PacC system so far described as the master control in the response to alkalinity.

## 2. Materials and Methods

### 2.1. Strains and Culture Conditions

*Aspergillus nidulans* was grown at 37 °C on solid complete medium (ACM) [46] to produce conidiospores for the inoculation of liquid cultures or propagation. Growth tests on solid media and liquid cultures for obtaining total RNA samples were performed using standard *Aspergillus* minimal medium (AMM) following [46]. Supplements, either salts or pH 8 buffer, were added when indicated in the text. Salts were either prepared as concentrated solutions or added directly to the AMM at the specified final concentrations. For the pH 8 condition, a final concentration of 100 mM Na_2_HPO_4_ was added.

The strains used in this work are indicated in Appendix A. These strains were obtained by either sexual crossing or transformation. For crossing, parental strains were point-inoculated on solid ACM and incubated for 48 h at 37 °C. Next, three hyphal contact areas of the parental strains were sectioned and placed on Petri dishes containing solid AMM lacking one of each parental strain requirements and containing 10 mM NaNO_3_ or a mixture of 10 mM NaNO_3_ and 5 mM ammonium tartrate as nitrogen sources. The plates were incubated for 2 days at 37 °C and then taped and further incubated for 15 days to induce sexual reproduction and the production of fruiting bodies (cleistothecia). Cleistothecia, 3 per cross, were selected on the basis of bigger size and spherical shape and cleaned of mycelial debris by rolling on AMM. Then ascospores were mechanically released from cleistothecia in 1 mL of sterile distilled water. This spore solution was inoculated on AMM agar plates and incubated for 48 h at 37 °C. Among the progeny, several colonies were selected and checked for the desired genotype. Crosses are indicated in Appendix A. The other two strains were obtained by protoplast transformation following the standardized protocol described in [47]. The strain MAD7634 expressing the PalF::HA_3_::Ub chimera was constructed by the transformation of MAD7631 with plasmid p1861 as described in [48]. To generate strains expressing the chimaera VPS23-GFP, a PCR fragment using oligonucleotides Vps23-GSP1: 5′-CAACAGATTGCTCCAGCACAGC and Vps23-GSP4: 5′-CCTGCACCGAATGTGCAACTATAC was generated from the genomic DNA of strain MAD3367 [22]. This PCR fragment contains the original construct comprising the 1.5 kb 3′ coding end of *vps23/AN2521* gene fused to *gfp/pyrG^Af^* marker and the 1.5 kb fragment of terminator [49]. Strains MAD7669 and MAD7670 were generated after the transformation of MAD5736 and MAD3919, respectively, with this PCR fragment.

### 2.2. RNA Extraction and Quantitative PCR

Mycelium from each strain was obtained by inoculating conidiospores into 250 mL flasks containing 50 mL of AMM supplemented with carbon (1% D-glucose) and nitrogen (5 mM ammonium tartrate) sources and with specific-strain requirements. Alkalinisation stress was induced by adding 100 mM Na_2_HPO_4_ to the culture and raising the pH from approximately 4 to 8 [1,43]. For the high-sodium cation assay, 1 M NaCl was added to the culture media. Mycelia were collected after 15, 30, 60 and 120 min of stress induction [44]. Mycelia were collected by filtration through Miracloth, dried and frozen for total RNA extraction following the TRIreagent protocol (Sigma-Aldrich, Merck Life Science S.L.U., Madrid, Spain). Total RNA extracts were then treated with Deoxyribonuclease I Amplification Grade (Invitrogen, ThermoFisher Scientific, Carlsbad, CA, USA) and SuperScript First-strand Synthesis System for RT-PCR (Invitrogen, ThermoFisher Scientific, Carlsbad, CA, USA) kits in order to obtain cDNA for quantitative PCR (Roche Lightcycler96, Roche Diagnostics S.L., Barcelona, Spain). The methodology was followed as described in [44]. The primers used for each gene analysed are listed in the primer table. All oligonucleotides have been designed with similar length, thermodynamic properties and 3′-terminal region of the gene of interest.

### 2.3. Protein Extraction and Immunodetection

For detecting tagged proteins in control conditions and after the induction of alkaline stress as well in the different strains studied, total protein extracts were obtained from filtered and lyophilised mycelia using the alkaline lysis protocol as described in [44]. For immunodetection of MYC_3_-tagged PacC (PacC900) [26], monoclonal mouse anti-c-MYC (1:10,000, clone 9E10, Sigma-Aldrich, Merck Life Science S.L.U., Madrid, Spain) was used. The forms of PalF (PalF500), and its chimera PalF-Ub, carrying 3 copies of haemagglutinin (HA_3_) [48] were detected using monoclonal rat anti-HA (1:1000, clone 3F10, Sigma-Aldrich, Merck Life Science S.L.U., Madrid, Spain). Polyclonal mouse anti-GFP (1:5000, clones 7.1 and 13.1, Sigma-Aldrich, Merck Life Science S.L.U., Madrid, Spain) was used for PacX-GFP detection with Western blot. The loading control used was α-tubulin protein, which was detected by monoclonal mouse anti-α-tubulin (1:5000, clone DM1A, Sigma-Aldrich, Merck Life Science S.L.U., Madrid, Spain). For all tagged protein detection described in this paper, peroxidase-conjugated goat anti-mouse immunoglobulin G (IgG, 1:4000; Jackson ImmunoResearch Laboratories, Cambridgeshire, UK) and goat anti-mouse IgG + IgGM (1:4000, Southern Biotech, bioNova cientifica S.L., Madrid, Spain) were used as secondary antibodies. Chemiluminescence detection in Western blot was achieved using the ECL kit (GE Healthcare, Madrid, Spain) and the ChemiDoc Imaging System implemented with Image Lab Touch software (version 2.2.0.08, Bio-Rad Laboratories, Madrid, Spain). Image processing and protein level measurements were treated using Image Lab (version 6.0, Bio-Rad Laboratories, Madrid, Spain) and CorelDRAW Graphics Suite X7.6 software.

### 2.4. Fluorescence Microscopy

Samples used in epifluorescence microscopy were prepared by inoculating conidia in 300 μL of Watch minimal medium (WMM) [50] in μ-slide 8-well uncoated (Ibidi GmbH, Inycom, Zaragoza, Spain), which was supplemented with 5 mM ammonium tartrate, 0.1% glucose, 25 mM NaH_2_PO_4_ and strain-specific auxotrophic requirements and incubated at 25 °C for 16 h. For alkaline pH stress assays, the standard culture media containing the grown mycelia were replaced for fresh media supplemented with 100 mM Na_2_HPO_4_. 

A Leica DMI-6000b microscope coupled to an ORCA-ER digital camera (Hamamatsu Photonics, Hamamatsu City, Shizuoka, Japan) and equipped with a 63 Plan Apochromat 1.4 N.A oil immersion objective (Leica Biosystems, Barcelona, Spain) was used to visualise the samples both in differential interference contrast (Normarski optics, DIC) and fluorescence detection [30]. Images were taken under standard conditions (Control) and, after pH alkaline stress induction, Vps23-GFP was visualised from initial times and PacX-GFP detection images were acquired up to two hours [22,31]. Metamorph (Molecular Dynamics, San Jose, CA, USA) software version 7.10.5.476 was used for image acquisition. For Vps23-GFP visualisation, it was necessary to perform a sum projection of a time-lapse stack. All images or stack images were processed using Image J Version 1.54 free software (https://imagej.nih.gov/ij/, accessed on 21 March 2024).

### 2.5. Statistical Analyses

For quantitative PCR experiments, samples were analysed in triplicate. The relative expression levels obtained were analysed using GraphPad v.8. For the expression profiles of each gene, data from each strain and condition were treated independently using a one-way ANOVA test with Tukey’s multiple comparisons test. For gene expression analyses of the *pacX20* and *sltAΔ* mutants, data from the pH conditions were analysed separately using the same tests as above. *p*-values are indicated by asterisks in GraphPad format: *, *p* < 0.01–0.05; **, *p* < 0.001–0.01; ***, *p* < 0.0001–0.001; and ****, *p* < 0.00001. Not significant *p*-values (≥0.05) are not shown in the graphs.

## 3. Results

### 3.1. Absence of SltA Function Altered pacC and palF Expression Levels in Response to Alkaline Ambient pH and High Sodium Concentration

In a previous transcriptomic analysis of the response of *A. nidulans* to an elevated extracellular sodium concentration and medium alkalinisation, we showed that SltA could play an important role on the expression levels of *pacC* and *palF* genes [44]. To extend these studies, we carried out a more detailed analysis of the expression patterns of the *sltA* and *sltB* genes, which comprise the Slt pathway, and the ambient pH-responsive *pacC*/*pal* pathway in response to these environmental stresses. Previous results from Northern, RNAseq and qPCR experiments showed a positive effect of ambient pH alkalinisation on the expression levels of *sltA* and *pacC*, but negative for *palF* [31,43,44]. Here, we determined the expression profiles of *sltA, sltB, pacC* and *palF* from mycelial samples of wild-type and null *sltA* strains grown under acidic (control) and then transferred to medium containing 1 M NaCl or buffered at an alkaline pH (hereafter referred to as pH 8) for 15, 30, 60 and 120 min.

Figure 1A shows the expression profile of the *sltA* and *sltB* genes in a wild-type strain at pH 8. Both genes showed an increase in expression levels upon alkalinity induction, a result consistent with previous data [44]. The expression of *sltA* increased within the first 15 min after shifting to pH 8 and reached a maximum expression after 30 min, followed by a decrease to basal levels. Similarly, *sltB* was up-regulated after alkalinisation, reaching its maximum expression at the 15 min time point and then fluctuating in expression levels close to those observed in the control condition. High extracellular sodium concentration had a different effect; *sltA* expression increased at 15 min at pH 8 but maintained this level also at 30 min after the shift, and slowly decreased transcription to initial levels at 60 and 120 min (Appendix A, left). The expression of *sltB* increased to a maximum at 30 min, at higher levels than those found at 15 min in the pH 8 condition (Appendix A, right). As reported previously by Northern assays and RNA sequencing [43,44], *sltB* expression was minimal in the absence of SltA function in both stress conditions [44].

We then measured the expression profile of *pacC*, which showed an up-regulation at pH 8 for 30 min and then returned to the initial expression levels after 120 min (Figure 1B). The profile showed the characteristic temporal up-regulation of *pacC* expression after the alkalinisation of ambient pH [31,44]. In contrast, in the null *sltA* background, the positive effect of ambient alkaline pH was attenuated and *pacC* showed a moderate expression peak after 60 min of shifting from ambient pH to alkalinity (Figure 1B, right). 

*palF* is the only pH-regulated Pal gene of the PacC signalling pathway [31]. The arrestin-like PalF is preferentially expressed at acidic pH and its function is essential for the signalling and correct processing of PacC^72kDa^ to PacC^53kDa^ [21,31,48]. The expression of *palF* in a wild-type strain under alkaline conditions was reduced over time (Figure 1C), showing the expected negative effect of alkaline ambient pH [31,44]. In a null-*sltA* strain, *palF* transcription did not show the down-regulation profile found in the wild-type, with no significant differences to those found in acidic conditions during the first 60 min after the pH shift (Figure 1C, right).

Sodium stress had different effects than pH 8 on *pacC* and *palF* expression. The expression levels of *pacC* were slightly increased after the addition of 1M NaCl (Appendix A) but *palF* expression showed a strong increase after 30 min incubation (Appendix A). The absence of *sltA* function almost blocked this response of the *pacC* and *palF* genes to 1 M NaCl (Appendix A).

These results reinforce our regulatory model predicting the existence of a temporally regulated expression of *sltA* and *pacC* when alkalinisation occurs [44], as well as in some elements belonging to these regulatory systems, and a possible involvement of SltA in maintaining the proper expression levels of *pacC* and *palF*. This led us to investigate the role of SltA in regulating the PacC/Pal pathway in response to alkaline pH.

### 3.2. Absence of SltA Activity Prevents the Proper Signalling of PacC and PalF upon Exposure to Alkalinity

To study PacC and PalF in a null *sltA* background, we constructed, by crossing a series of *sltAΔ* strains expressing fully functional tagged forms of either PacC or PalF, MYC_3_-PacC and PalF-HA_3_, respectively (Appendix A) [21,26]. Consistent with previous studies in complex fermentation media [25], samples from cultures using minimal medium (AMM) showed that the primary 72 kDa form of PacC (PacC^72kDa^) was rapidly signalled to the 53 kDa form (PacC^53kDa^) upon increasing environmental pH, and then processed to the active 27 kDa form (PacC^27kDa^) in the wild-type strain (Figure 2A). After this sequential proteolytic processing of PacC, the recovery of the 72 kDa primary form was observed after 120 min of alkaline induction (Figure 2A), while the 53 kDa intermediate form and the 27 kDa active forms tended to accumulate. In the absence of *sltA*, we observed a stronger signal of PacC^72kDa^ under acidic conditions (Figure 2A, lanes “C”). 

Compared to the previously well-established pattern of PacC proteolytic processing in response to alkalinity in the wild-type, the absence of SltA function affected this pattern (Figure 2A). The PacC^72kDa^ form was detected throughout the experimental time course, but its signal decreased after 1 h of induction at pH 8. The intermediate 53 kDa form was barely detected. The proteolysed 27 kDa form of PacC is, in fact, a set of bands that correspond to alternative sites of proteolytic processing by the proteasome [27]. In these experiments, bands corresponding to PacC^27kDa^ accumulated after the shift to pH 8 as in the wild-type, but we observed a stronger signal from a PacC^27kDa^ form with the lowest mobility compared to that detected in the wild-type (red arrow, Figure 2A).

In an attempt to detect PacC^53kDa^ and the correct processing of PacC^27kDa^ in the *sltA* null background, we analysed longer incubation periods under pH 8 condition and also used a medium combining alkalinisation with calcium supplementation, since extra calcium suppresses some phenotypic defects of the *sltAΔ* mutant [43,44]. However, none of these conditions suppressed the altered proteolytic processing of PacC in the absence of SltA (Appendix A). Calcagno and collaborators [51] examined the proteolytic processing of PacC in strains carrying different mutant alleles of *sltA*. Among the mildest *sltA* loss-of-function mutants isolated in that work were *sltA59* and *sltA60* alleles, both of which truncate SltA at amino acid 607 of 678. In our growth conditions, we studied the proteolytic processing of PacC in the *sltA60* background. Interestingly the absence of the last 71 amino acids of SltA affected PacC signalling. Although PacC^72kDa^ was still visible after 15 and 30 min of the shift to pH 8 (compare blots in Appendix A), we detected the intermediate PacC^53kDa^ form and the partial recovery of the correct processing of PacC^27kDa^. These results reflect that although there is no reduction in protein levels corresponding to the lower expression of *pacC* in a mutant *sltA* background, there is an atypical pattern of PacC protein signalling and processing that could also affect the function of its active 27 kDa form, resulting in an inability to respond to an alkaline environment.

To investigate a possible effect on the signalling process of PacC, we studied the expression of PalF and its post-translational modifications (PTMs) in the absence of SltA function at an alkaline pH. In the wild-type, PalF protein was detected with similar intensity in all samples and, consistent with ubiquitination and phosphorylation of PalF following alkaline pH signalling [21,31], resulted in the visualization of low mobility forms, mainly at 15 and 30 min after treatment (Figure 2B, see also the *pacX20* background and the PalF-Ub chimera). In the null-*sltA* background, the primary form of PalF was detected in a similar proportion to that found in the wild-type. However, the PalF PTMs observed in the wild-type at 15 and 30 min after alkalinisation were not detected in the null *sltA* background even at longer exposure times (Figure 2B).

The overall data support the view that SltA is involved in the correct expression levels of the *pacC* and *palF* genes, but also in the post-translational modifications that occur on PacC and PalF. The absence or severe reduction in phosphorylated/ubiquitinated forms of PalF could affect the correct processing of PacC^72kDa^ to PacC^53kDa^ and then to PacC^27kDa^, preventing an effective response to environmental alkalinity.

### 3.3. Absence of the Suppression of Null-sltA Phenotype by a pacX Null-like Mutation

The absence of PacX activity results in higher pH-independent expression of *pacC* [31]. We combined the *pacX20* mutation, which mimics a *pacX* null allele, with different mutations in the Slt system to investigate a possible link between the absence of SltA and PacX activities. Figure 3A shows that strains carrying combinations of *pacX20* and *sltA* or *sltB* null mutations did not show a major reduction in colony growth on AMM. These double mutants showed poor sporulation compared to the wild-type strain or the single mutants (Figure 3A, AMM column). When tested at different cation and pH stresses, the *pacX20* strain grew as well as the wild-type control strain. Null *sltA*, null *sltB* and *sltA1* mutants exhibited the expected sensitivity to high concentrations of NaCl, LiCl or alkaline pH (Figure 3A). The *sltA1* mutation truncates SltA after the zinc finger domain and results in a non-functional SltA protein [36,43]. Deletion of *sltB* prevents the processing of full-length SltA^78kDa^ into the 32 kDa form (SltA^32kDa^) [43]. Double mutants behaved indistinguishably from single *sltA* or *sltB* mutants, demonstrating that the absence of PacX function does not rescue the absence of SltA.

To investigate the dependence of *pacX* expression on SltA function, we quantified *pacX* transcript levels using qPCR under acidic conditions and after the transfer to alkaline pH in both wild-type and mutant strains. Our results showed strong variability in *pacX* expression levels in the wild-type strain after the shift to alkaline conditions (Figure 3B). We observed a significant increase in *pacX* expression only after 15 min at an alkaline pH. When analysing the *pacX* expression profile in the *sltAΔ* mutant, a significant difference was observed after 30 min compared to the control condition (Figure 3B). The *pacX* levels were analysed after 60 min of exposure to alkaline pH in the *pacX20* mutant background and in the double *pacX20 sltAΔ* mutant, revealing significant differences between them (Figure 3C). In comparison to the levels found in the *pacX20* background, the expression levels of *pacX* were significantly reduced by the presence of the *sltAΔ* mutation, regardless of the ambient pH (Figure 3C).

We also used a PacX-GFP tagged form [31] to assess the effect of alkalinisation and absence of SltA function on PacX protein levels and cellular localization. In contrast to mRNA levels (Figure 3B), PacX protein levels decreased from those measured at acid pH when transferred to alkaline pH but remained essentially constant for 60 min at pH 8 (Appendix A). In the absence of SltA an increased signal of an intermediate form of PacX was visible, and the primary form slightly increased its signal after a 60 min transfer to pH 8 (Appendix A). As previously reported [31], alkalinisation did not alter the specific subcellular localization of PacX-GFP, which mainly accumulated in a nuclear punctate location. No changes in the subcellular localization of PacX were observed in the *sltA* null background at either an acid or alkaline pH (Appendix A). 

Thus, an epistatic effect of *sltA* over *pacX* was observed in phenotype assays under stress conditions and in expression studies where the activity of SltA is required for the proper expression of *pacX* at alkaline pH.

### 3.4. Absence of PacX Activity Did Not Rescue PacC and PalF Transcriptional and Translational Profiles in a sltAΔ Background

As PacX acts as a negative regulator of *pacC* expression, we studied the relative expression of *pacC* and *palF* in *pacX20*, null-*sltA* and the double mutant backgrounds. As a control, qPCR data in Figure 4A,B show the transcriptional response of *pacC* and *palF* to alkalinity, which is consistent with previous data from Northern analysis showing that *pacC* is an alkaline-expressed gene and *palF* is an acid-expressed gene (see also Figure 1B,C). As described by Bussink and collaborators [31], *pacC* expression increased in the absence of PacX function under both acid and alkaline pH conditions. However, the double mutant *sltAΔ pacX20* exhibited expression levels similar to the wild-type under acid pH. Under alkaline conditions, *pacC* expression increased to a level higher than that observed in the *sltA* null mutant but lower than that detected in the wild-type (Figure 4A). 

Down-regulation of the *palF* expression was observed at pH 8 in both the wild-type strain and in the null *sltA* mutant. In the *pacX20* background, *palF* expression levels were lower in both conditions but maintained a differential expression according to the environmental condition (Figure 4B). Loss of SltA and PacX activities caused a reduction in relative expression to its minimum level, whereas alkalinisation caused an increase in *palF* expression.

Higher levels of *pacC* expression led to the detection of larger amounts of both full length and processed forms of PacC, even in the absence of PacX activity (Figure 4C and Appendix A). See also [31]). In the single null *sltA*, the processing of PacC^72kDa^ was strongly delayed when the ambient pH was shifted to 8. The intermediary PacC^53kDa^ form was barely detected, and we observed the accumulation of an anomalous low mobility version of the PacC^27kDa^ form (Figure 2A and Appendix A). The presence of the *pacX20* mutation in the *sltA* null strain did not result in any changes in the processing pattern, and the PacC^53kDa^ form was not detected, despite an increase in the protein levels of PacC (Figure 4C and Appendix A). Additionally, altered PacC proteolytic processing was observed at pH 8 in the presence of null *sltB* or *sltA1* mutant alleles (Appendix A). The absence of SltB results in a non-proteolysed SltA^78kDa^ primary form and the SltA1 protein is a truncated version of SltA after the DNA binding domain (aa 501), resulting in a non-functional mutant protein [43]. Therefore, the presence of only SltA^78kDa^ and the truncated SltA(1-501) forms mimic the effect of the null *sltA* allele on PacC processing.

The *pacX20* mutation did not affect the expression (Figure 4B) or protein levels (Figure 4D) of *palF.* However, in the null *sltA* background, the PTMs of PalF were not detected at alkaline ambient pH. This suggests that proper signalling between PalF and the downstream elements of the Pal pathway may be affected. The process of PacC proteolytic processing initiates with the recruitment of the ESCRT-I protein Vps23 by ubiquitinated PalF (PalF-Ub). Following the pH shift signal, PalF ubiquitylation promotes the recruitment of Vps23^AN2521^ (Vps23 for brevity) through binding to the box 1 and box 2 regions in PalF (Figure 5A) [23,48]. To analyse whether Vps23 is properly recruited to plasma membrane accumulations at alkaline pH in the absence of SltA, we obtained a sltAΔ vps23-GFP double mutant and conducted fluorescence microscopy. Figure 5B displays the localization of Vps23-GFP in three different backgrounds: a functional Pal pathway (wild-type strain), the mutant *palF15* and the newly generated *sltAΔ* strain. In the absence of stress, Vps23-GFP was uniformly detected in the cytoplasm of the wild-type strain. When the extracellular pH was increased (Figure 5B), several intense puncta located close to the plasma membrane were observed, indicating the interaction between Vps23 and PalF, as previously noted in [22]. To serve as a negative control, we used the *palF15* mutant. This mutant results in an early truncation of the protein at Leu189, which prevents the formation of these puncta at pH 8. As a result, the Vps23-GFP fluorescence remains dispersed along the cytoplasm (see also [22]). The absence of sltA did not modify the pattern observed in the wild-type strain. Environmental alkalinisation led to the characteristic Vps23 localization in puncta close to the plasma membrane in both the *sltA*Δ and wild-type strains.

Based on these results, the interaction between PalF and Vps23 is possible in the null *sltA* mutant, despite the low levels of the post-translationally modified PalF forms. This suggests that other elements downstream of Vps23 might be affecting PacC signalling and processing when SltA activity is absent.

### 3.5. Altered Proteolytic Processing of PacC^72kDa^ in a Null sltA Background Is Independent of the PalB Protease

The Pal pathway mediates the signalling of PacC^72kDa^ at ambient alkaline pH, requiring the function of PalA and PalB to recruit PacC^72kDa^ to the membrane and undergo proteolytic processing to PacC^53kDa^, respectively [24,26,52]. In a null *sltA* background, the proper localization of Vps23-GFP was detected under both acid and alkaline pH conditions. We investigated whether Vps23 signalling facilitated the recruitment of the other Pal pathway components. A double mutant lacking the *sltA* and *palB* genes was obtained to study their effect on the signalling, and proteolysis of PacC^72kDa^ was affected. The phenotype test in Figure 6A showed an epistatic effect of *sltAΔ* over the *palBΔ* mutation in response to 1 M NaCl. Under the molybdate condition, we observed a synergic effect of both mutations. The colonies of the double mutant showed a compact growth similar to that observed in a single null *palB*, and a brown-caramel-like colour as in the single mutant *sltAΔ*. We then analysed the proteolytic pattern of PacC900 in the pH 8 condition in this *palBΔ sltAΔ* mutant.

Figure 6B shows an immunodetection of PacC900 where the absence of PalB function leads to an accumulation of PacC^72kDa^ at an alkaline pH and only a residual processing was detected in acid and alkaline ambient pH (Figure 6B, single asterisk). However, in the *palBΔ sltAΔ* mutant, the proteolytic processing of PacC in response to alkaline pH was similar to that observed in the single null *sltA*. Despite the absence of PalB function, we were able to detect an effect of medium alkalinisation. At an acid ambient pH, only PacC^72kDa^ was detected as in the single mutants and the wild-type strain. However, at an alkaline pH, both low and high electrophoretic mobility forms of PacC^27kDa^ were detected even in the absence of PalB function (Figure 6B, single and double asterisks). These results clearly establish that the anomalous proteolytic processing of PacC observed in the absence of SltA is independent of the PalB protease.

### 3.6. Constitutive Activation of the Pal Pathway Was Unable to Suppress sltAΔ Phenotypes

The low levels detected of post-translationally modified forms of PalF, as well as previous results from null *palB* mutants, suggest that an unstructured Pal pathway might be the cause of the anomalous processing of PacC in a *sltAΔ* background. To test this hypothesis, we generated a triple mutant *sltAΔ*, *pacC900* (MYC_3_-PacC) and *palF* fused to both HA3 and ubiquitin (Ub) tags (PalF-HA-Ub) and compared it to a similar strain expressing PalF500 (PalF-HA_3_). This constitutively ubiquitinated PalF form (PalF-HA_3_-Ub) has been shown to activate the Pal pathway independently of ambient pH. This results in the constitutive processing of PacC^72kDa^ [48] at acidic conditions (Figure 7A, top insert, lane 5, compare with control lanes 1 and 3). In the *sltAΔ palF-Ub* background, the primary form PacC^72kDa^ accumulated at an acidic pH but the PacC^27kDa^ form was also visible (Figure 7A, lane 6). The mobility of the fully processed PacC form in the *sltAΔ* mutant was similar to that observed in the wild-type, whether expressing PalF-HA_3_-Ub or not. This suggests that PacC is now correctly signalled (compare lanes 2, 4 and 6). In contrast, the intermediate form PacC^53kDa^ was not rescued to normal levels and remained almost undetected in a null *sltA* background. Nevertheless, the low amount of this form presented a similar electrophoretic mobility when compared to that found in the wild-type protein extract (Figure 7A, lanes 2, 5 and 6). The presence of the triple mutant of modified forms of PalF-HA_3_-Ub, ubiquitinated and probably the phosphorylated forms, was confirmed by using specific antibodies for HA.

Despite PalF-HA_3_-Ub being constitutively expressed, the accumulation of the primary form and low detection of the intermediate form is maintained in a *sltAΔ* background. In contrast, the accumulation of the active 27 kDa form seemed restored. We then analysed whether this change in PacC processing had any phenotypic effect on the *sltAΔ palF::ha_3_::ub* mutant. Figure 7B shows that the strain expressing the constitutively ubiquitinated *palF* construct was not phenotypically different from the wild-type on standard AMM medium or buffered at an alkaline pH. However, the constitutive activation of the Pal pathway resulting in the pH-independent accumulation of PacC^27kDa^ conferred an increased resistance to 5 mM sodium molybdate. The null *sltA* strain showed sensitivity to alkaline pH and the accumulation of a brown-caramel pigment when growing on medium containing sodium molybdate. The *sltAΔ palF::ha_3_::ub* mutant displayed the same phenotype as the *sltAΔ* strain for the conditions studied. The constitutive activation of the Pal pathway and the presence of a PacC^27kDa^ form with the expected mobility are not sufficient to bypass the pH-sensitive phenotype displayed by the *sltAΔ* mutation.

### 3.7. Epistasis of Null sltA Allele over pacC Constitutive Mutations

As the null *sltA* mutant may compromise the transmission of ambient pH signalling and be responsible for the absence of the suppression of the pH 8-sensitive phenotype, we generated strains where PacC proteolysis was independent of the Pal-pathway. The *pacC^c^14* mutation causes a truncation at amino acid 492, yielding a protein with a similar size to the intermediate PacC^53kDa^ form [18]. In the *pacC^c^14* mutant background, PacC activation only requires the pH-independent proteolytic step carried out by the proteasome to produce PacC^27kDa^. We also generated a double *sltAΔ pacC^c^14* mutant strain to determine the effect of the absence of SltA activity on pH-independent PacC processing. PacC^c^14 processing was visualized using the MYC3 tagged allele, *pacC^c^14900*. The strain expressing MYC3-PacC^c^14 exhibited a phenotype similar to that of the wild-type control strain when grown on media buffered at an alkaline pH or supplemented with calcium or molybdate at the concentrations shown in Figure 8A. In contrast, the double mutant *sltAΔ pacC^c^14* behaved similarly to the single *sltAΔ* mutant, displaying sensitivity to alkaline culture conditions (Figure 8A). Moreover, the double mutant showed morphological defects, such as reduced colonial growth and poor sporulation, in all conditions studied. The double mutant also lacked the tolerance to sodium molybdate described for *pacC^c^14* single mutants [18,48], at both concentrations tested. The PacC forms derived from the *pacC^c^14* mutation at an acid pH were visualized using the anti-MYC3 antibody (Figure 8B). In the *pacC^c^14900* background, both the PacC^53kDa^-like form and the 27kDa processed form were observed (Figure 8B, left, compare lanes 1 and 2, with 3 and 4 in the wild-type and 5 and 6 in the null *sltA* mutant). The presence of the *sltAΔ* allele did not affect the detection of the PacC^53kDa^-like form or the mobility of PacC^27kDa^ compared to the single *pacC^c^14900* mutant. Therefore, the absence of SltA did not alter proteasome activity on the PacC^53kDa^-like form, yet activation of PacC independently of the Pal pathway did not suppress the phenotypic effect of a null-*sltA* background.

*PacC^c^700* is a chimeric PacC construct displaying constitutive function [53]. The PacC700 protein mimics the PacC^27kDa^ form (PacC5-250) and carries GFP fused at the N-terminus. A strain expressing PacC700 shows an alkalinity mimicking phenotype [53]. The aim of using this construct was to determine whether constitutive expression of PacC^27kDa^ alters its protein levels and cellular localization in a *sltAΔ* background and whether this mutant exhibits a tolerance to alkalinity. The double mutant *sltAΔ pacC^c^700* was obtained by crossing. Colonies of this double mutant exhibited on AMM a stronger morphological defect than the single *pacC^c^700*, showing reduced colony radial growth and poorer conidiation (Figure 9A). When tested at pH 8, with two concentrations of sodium molybdate or sodium chloride, the double mutant displayed a similar phenotype to that of the single *sltAΔ.* Growth tests showed that the *sltAΔ* phenotype prevails over the *pacC^c^700* phenotype (Figure 9A).

Western blot analysis using an anti-GFP antibody to detect PacC700 did not reveal differences between the *pacC^c^700* mutant and the *sltAΔ pacC^c^700* double mutant. Neither alkaline growth conditions nor the *sltA* mutation caused major differences in GFP-PacC^27kDa^ signal (Figure 9B). Next, we evaluated the localization of PacC700. As previously described [53], the PacC700 protein shows continuous accumulation in all nuclei in a cell compartment. The presence of the *sltAΔ* allele did not affect this localization (Figure 9C). Hence, even though the cellular localization and protein amount remain unaltered in a null *sltA* mutant that constitutively expresses the active form PacC^27kDa^, the strain remains sensitive to stress environments caused by alkaline pH or a high cation concentration.

The overall results show that SltA activity is necessary for the correct response to alkaline pH. SltA may also participate in the regulation of other pathways, in addition to modulating PacC/Pal pathway functionality.

## 4. Discussion

The ability of microorganisms to survive under unfavourable environmental conditions depends on the generation of proper transcriptional and/or post-translational responses. At least three major regulatory circuits drive tolerance to environmental alkaline pH in *Aspergillus nidulans* [1]. This study aims to enhance our comprehension of the coordination between two systems and demonstrates that the activity of the transcription factor SltA is required for optimal functioning of the Pal/PacC system. Calcagno and collaborators previously reported potential connections between these regulatory pathways by linking the Slt system and the activity of some of the Vps proteins, which are part of both the intracellular trafficking machinery in ESCRT complexes and the PacC/Pal signalling pathway [51,54,55,56]. Complete or partial loss-of-function mutations in the Slt system rescued the debilitating growth phenotype of several *vps* null mutants, indicating a role of SltA and SltB in the intracellular trafficking of proteins [43,51,57]. Despite the wide knowledge on the Pal/PacC pathway, only PacX and PacC itself have been identified as negative and positive regulators of *pacC* expression [29,31]. Recently, Picazo and collaborators described an effect of the lack of SltA activity over *pacC* expression [44].

Previous work analysing the expression level of genes in the *A. nidulans* PacC/Pal pathway involved the use of Northern techniques. This study presents quantitative profiles of *pacC*, *palF* and the *slt* genes under alkaline pH and high sodium concentration. The results demonstrate a common response to alkalinity, with increased expression levels of both *pacC* and *sltA*. As expected, *pacC* showed the known up-regulation during early exposure to alkaline pH, while the opposite was observed for *palF*, the only *pal* gene under pH regulation [31]. In contrast, the transcriptional response to sodium stress differs from that of alkaline pH. The previously observed inverse regulation of *pacC* and *palF* was not detected under high sodium concentration, consistent with the absence of genetic and phenotypic results involving the PacC/Pal pathway in salt tolerance (compare Figure 1A and Appendix A). In contrast, the transcriptional profiles of *sltA* and its partner *sltB* were distinct in each stress condition. The dual SltA/B system is coordinated at the transcriptional level where *sltB* levels are inversely regulated by *sltA* levels (Figure 1A and Appendix A).

The absence of SltA activity (null allele) not only affects high level *sltB* expression but also a significant number of genes [44]. The transcriptional profiles of *pacC* and *palF* are also altered. PacX is the only known trans-regulator of *pacC* mRNA and protein levels. We analysed whether PacX could coordinate its activity with SltA. Our results showed that SltA function is necessary to maintain *pacX* transcription and the up-regulatory effect of lacking PacX is also mediated by SltA (Figure 3B,C and Figure 4A). The absence of PacX and SltA function does not up-regulate *pacC* or *palF* expression levels. However, the effect on protein levels is still observed (Figure 4 and Appendix A). This result indicates that the lack of PacX function is not solely related to transcription, but also affects the translation of *pacC* and *palF* mRNAs. The elevated concentrations of PacC protein generated by the increase in transcription explained the suppression by *pacX* mutant alleles of phenotypes in partial loss-of-function *pacC* mutants [31]. Here, we show that this effect in elevating protein expression levels by *pacX* mutants does not reverse the phenotypes caused by the absence of SltA, including the anomalous proteolytic processing of PacC.

The impact of SltA on the PacC system extends beyond the transcriptional regulation. In the null *sltA* strain, the proteolytic pattern of PacC activation is completely altered. Alkaline pH signalling of the primary form of PacC, PacC^72kDa^, is almost blocked in the absence of SltA. Additionally, the intermediate PacC^53kDa^ is barely detectable, and the fully proteolysed form of 27kDa shows anomalous electrophoretic mobility (Figure 2, Figure 4C, Appendix A). PacC^53kDa^ has been linked to the peak of *pacC* expression [25] as this form is almost predominantly present in the fungal cell at the early times of exposure to alkalinity. However, it has not been proven, as there is no way to mimic such intermediate form through reverse genetics. PacC^53kDa^ is committed to proteolysis by the proteasome [27]. The lack of a positive effect of alkaline pH on *pacC* transcription in the *sltA* null background correlates with the extremely low levels of PacC^53kDa^. This strongly supports the role of the intermediate form of PacC in its autoregulation.

PTMs in PalF are also affected in the *sltA* null mutant. The low detection of modified forms of PalF, caused by phosphorylation and ubiquitination, in the absence of SltA indicates that Pal signalling is affected. Two strategies were used to demonstrate the role of these PTMs in PalF. Our results indicate that the recruitment of the additional elements of the pathway to static foci at the plasma membrane is unaffected by the absence of SltA (Figure 5B), suggesting that the reduction in these forms in this mutant background is not sufficient to prevent their recruitment or that other regions in PalF are involved in these interactions. The use of a PalF allele expressing a constitutively ubiquitinated form of PalF partially restores PacC processing in the *sltA* null background. This result shows the importance of ubiquitination in the proper processing of PacC, as previously noted [22,48]. Additionally, it suggests a defect in the ubiquitination machinery in the null *sltA* mutant (Figure 7A). We have examined the transcriptional data on the known elements of the mono and poly-ubiquitination machinery without finding convincing evidence of a defective mechanism (RNAseq data from [44]). Therefore, we hypothesize that the absence of SltA could interfere with the appropriate signalling of PalF either by hindering the recruitment of PTM machinery or by negatively affecting any of the proteins involved in this machinery. 

The anomalous mobility of the PacC^27kDa^ form in the null *sltA* background is comparable to that observed in the proteasome mutants [27]. The proteolysis of PacC^53kDa^ to PacC^27kDa^ by the proteasome results in several forms, depending on the conformations and PTMs adopted by PacC during its passage through the proteasome [27]. This study demonstrates that the pattern of PacC^27kDa^ forms in the null *sltA* is independent of its protease PalB, which is required in the wild-type background to generate the intermediary form PacC^53kDa^. It is postulated that the PacC^72kDa^ form undergoes a transition between open and close conformations [29,58]. This transition could account for the formation of a PacC^27kDa^ form with low electrophoretic mobility. The open PacC^72kDa^ form may serve as an alternative substrate for the proteasome. The efficiency of proteolytic processing by the proteasome must be significantly lower, but still sufficient to generate the accumulation of a specific (low mobility) PacC^27kDa^ form at alkaline pH. Nevertheless, this form must be partially functional. Sodium transporters encoded by *enaA* and *enaB* genes are required for growth at alkaline pH and are tightly regulated by PacC [30]. Since the expression of *enaA* and *enaB* genes is not affected in the absence of SltA [30], we predict that such low mobility in the PacC^27kDa^ form is functional or that the low levels of PacC^27kDa^ of higher electrophoretic mobility are sufficient for the transcriptional regulation of these genes.

An important finding of this genetic analysis is that null mutations in the Slt pathway are epistatic to *pacC* gain-of-function mutations. We found that the ambient pH-sensitive phenotype of null *sltA* strains is not suppressed, despite the recovery of the correct PacC^27kDa^ form in the null *sltA* background either by introducing a PalF constitutive ubiquitination or by using mutant *pacC* alleles independent of the Pal signalling pathway. It is intriguing why stains carrying Slt- mutations were not isolated during the extensive screenings for the suppressors of *pacC* or *pal* mutants [18,29]. A possible answer is that *slt*- mutants exhibit a tolerance to molybdate anion and sensitivity to neomycin (this work, [38]); both phenotypes are incompatible with the expected phenotypes of the suppressors of *pacC* and *pal* gain-of-function mutations, which should exhibit sensitivity to molybdate and resistance to neomycin [18]. The fact that the absence of SltA function does not prevent the expression of *enaA* and *enaB* genes [30] strongly supports a new model of regulation in which SltA participates in ambient pH-responsive pathways different from those in which PacC is involved. This work establishes, for the first time, the hierarchy between two key ambient pH regulatory systems in fungi (Figure 10), the wide phylogenetically distributed and well-studied PacC/Pal system and the Pezizomycotina specific Slt system.

## Figures and Tables

**Figure 1 cells-13-00651-f001:**
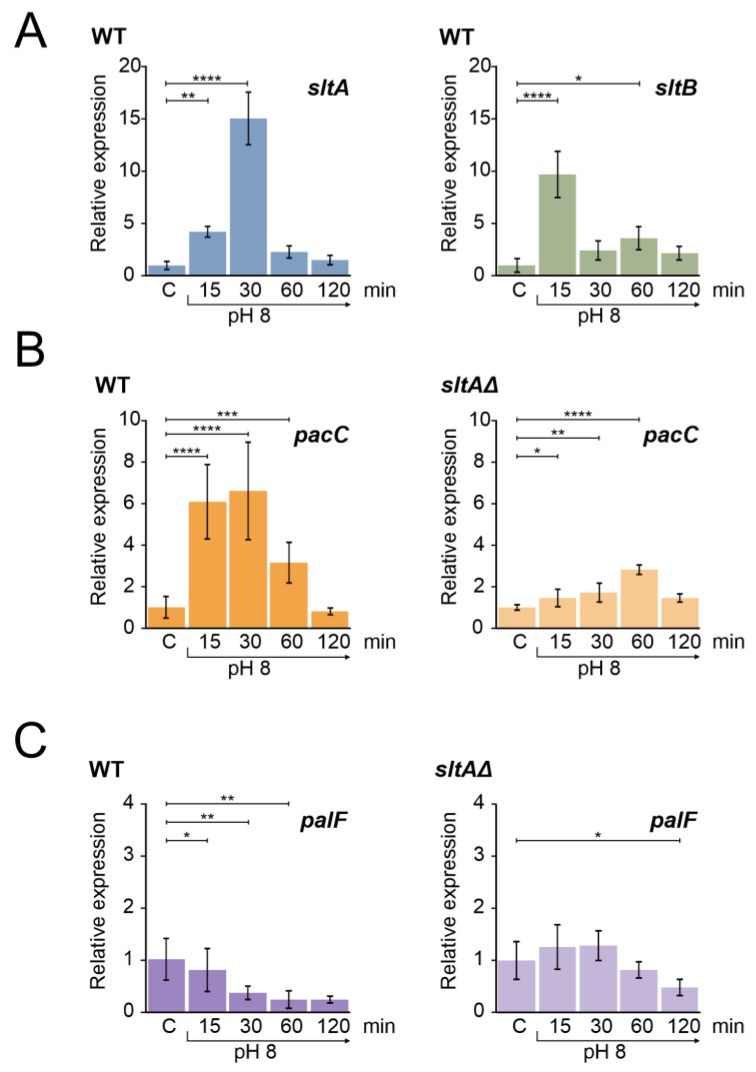
Gene expression profile of genes from *slt* and ambient pH regulatory pathways in wild-type and *sltAΔ* strains under alkaline pH condition. (**A**) Expression levels of *sltA* and *sltB* in the wild-type strain MAD3652 (WT). (**B**) Expression of *pacC* in the WT and *sltAΔ* strain (MAD3816). (**C**) Expression levels of *palF* in the WT and *sltAΔ* strain. Mycelia were grown in AMM in independent samples for each experimental time. For alkalinisation of extracellular pH, 100 mM Na_2_HPO_4_ were added to culture media. Mycelia were collected after 15, 30, 60 and 120 min. Relative expression data were normalised using the control condition expression levels of each strain. The β-tubulin-encoding gene *benA/An1182* was used as a reference gene. Error bars represent the standard deviations of three replicates for each sample result. *, *p* < 0.01–0.05; **, *p* < 0.001–0.01; ***, *p* < 0.0001–0.001; ****, *p* < 0.00001. Not significant *p*-values (≥0.05) are not indicated in graphs.

**Figure 2 cells-13-00651-f002:**
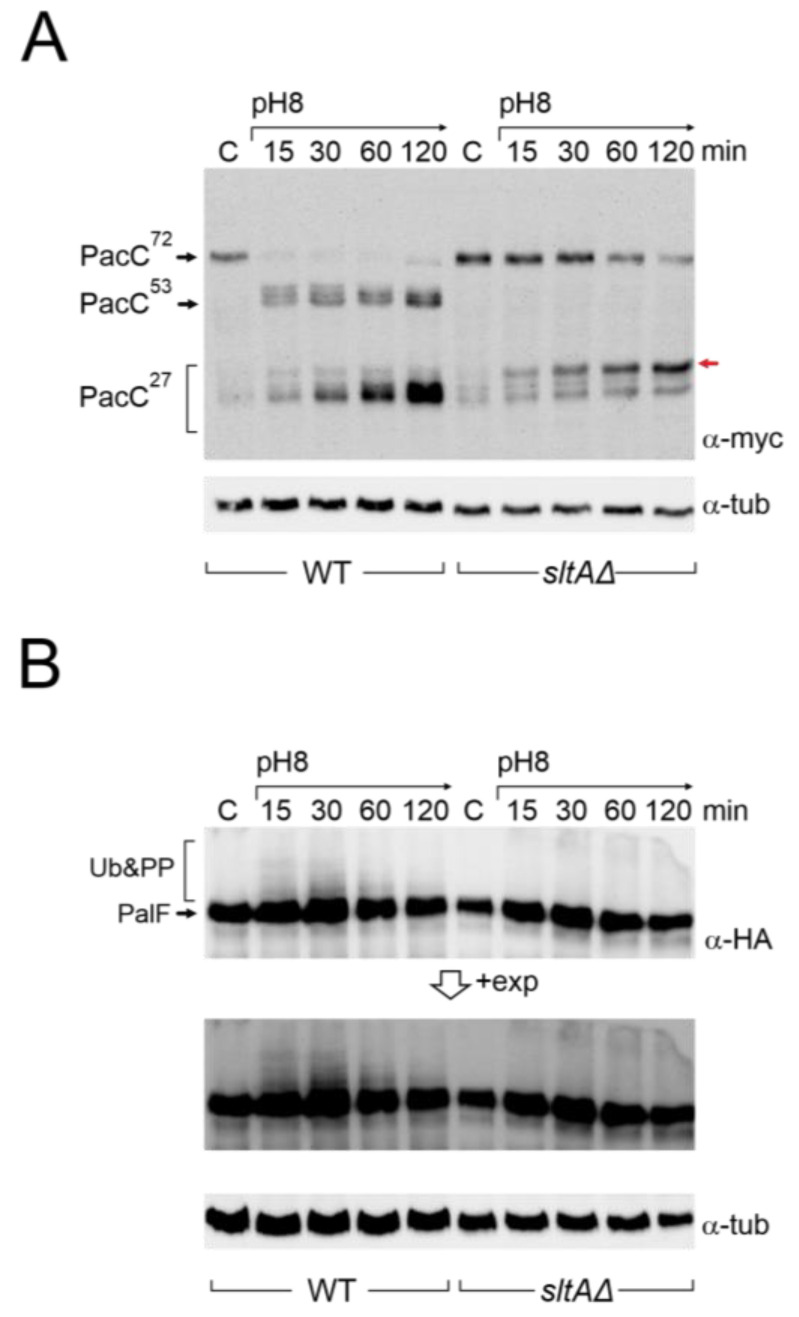
Immunodetection of tagged forms of PacC and PalF proteins under alkaline pH in wild-type and *sltAΔ* strains. (**A**) PacC900 (MYC_3_-PacC protein) was detected using specific anti-myc antibody (α-myc) in total protein extracts from wild-type (MAD7627) and *sltAΔ* (MAD7630) strains. Red arrow indicates the low mobility band for PacC full proteolysed version (**B**) Detection of PalF-HA-tagged protein (PalF500) using a specific anti-HA antibody (α-HA). Detection of α-tubulin was used as a loading control. A highly exposed image (+exp) was taken for a better visualisation of PalF post-translational modifications (Ub&PP) at alkaline pH conditions.

**Figure 3 cells-13-00651-f003:**
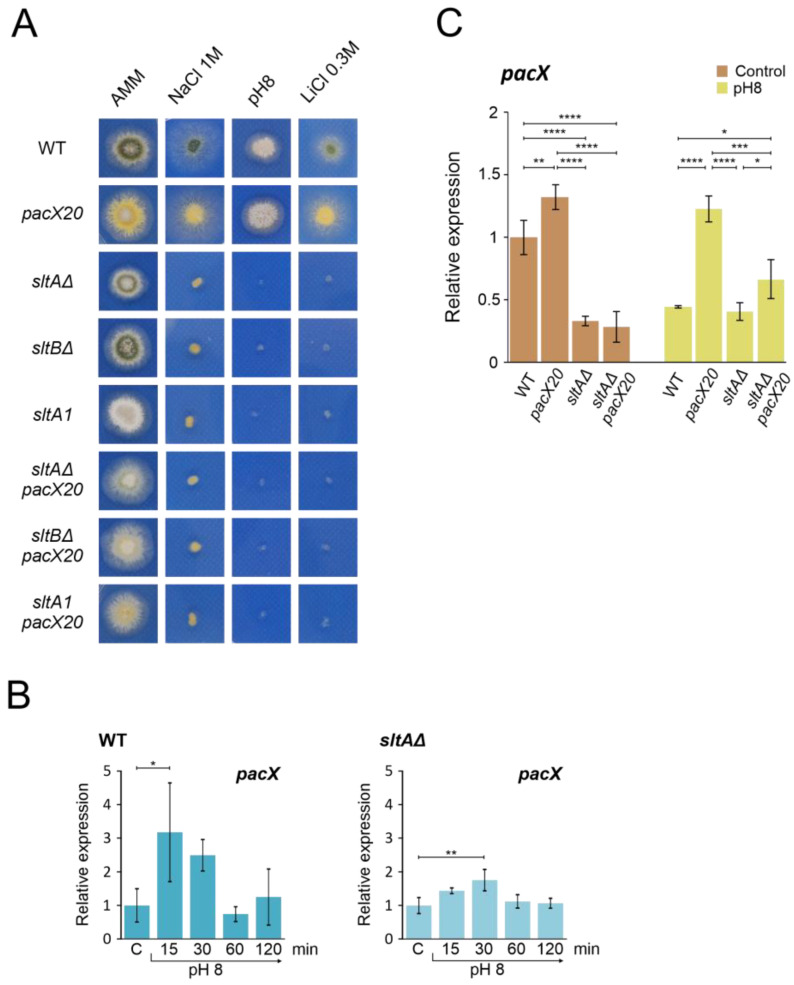
Epistatic effect of *slt*- mutations over *pacX20* mutation. (**A**) Phenotypic analysis of strains combining the absence of PacX function by a *pacX20* allele and null alleles of *sltA* and *sltB*, or the complete loss of function allele *sltA1*. Growth of single *pacX20* strain (MAD1652) did not show differences compared to wild-type strain (MAD6669) under 1 M NaCl, pH 8 (100 mM Na_2_HPO4) and 0.3 M LiCl. Double mutant strains *sltAΔ pacX20* (MAD7621), *sltBΔ pacX20* (MAD7623) and *sltA1 pacX20* (MAD7622) displayed similar phenotype as the single *slt*- mutants, *sltAΔ* (MAD3816), *sltBΔ* (MAD3693) and *sltA1* (MAD1132) under stressful conditions. Strains were grown at 37 °C for 48 h and colonial growth was imaged. (**B**) Transcriptional profile of *pacX* gene in wild-type (WT) and *sltAΔ* strains under alkaline pH stress conditions. (**C**) Relative expression levels of *pacX* in mycelia of WT and single and double *sltAΔ pacX20* mutant strains at alkaline and acid (Control) ambient pH. The *benA* gene was used as reference gene and relative expression level was normalised according to control condition of wild-type strain. Error bars represent the standard deviations of three replicates for each sample result. *, *p* < 0.01–0.05; **, *p* < 0.001–0.01; ***, *p* < 0.0001–0.001; ****, *p* < 0.00001. Not significant *p*-values (≥0.05) are not indicated in graphs.

**Figure 4 cells-13-00651-f004:**
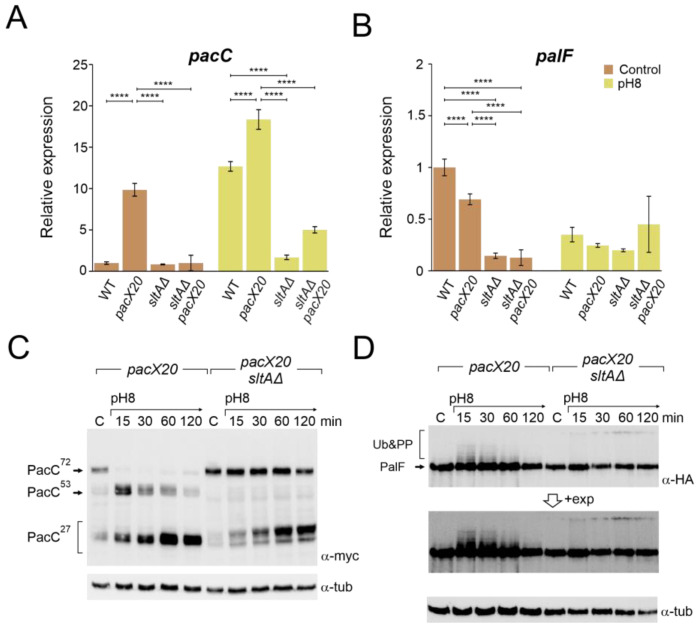
Absence of PacX function does not rescue *pacC* or *palF* transcriptional levels or protein patterns in a null *sltA* background. Expression levels at pH 8 of *pacC* (**A**) and *palF* (**B**) in the double mutant *pacX20 sltAΔ* background (strains used as in Figure 3B). The β-tubulin *benA* gene was used as reference to measure relative expression. Visualization by immunodetection of changes in the proteolytic patterns of MYC and HA-tagged proteins PacC900 (**C**) and PalF500 (**D**) in a single *pacX20* (MAD7628) and the double mutant *pacX20 sltAΔ* (MAD7629) strains. Detection of α-tubulin protein was used as the loading control. Error bars represent the standard deviations of three replicates for each sample result. ****, *p* < 0.00001. Not significant *p*-values (≥0.05) are not indicated in graphs.

**Figure 5 cells-13-00651-f005:**
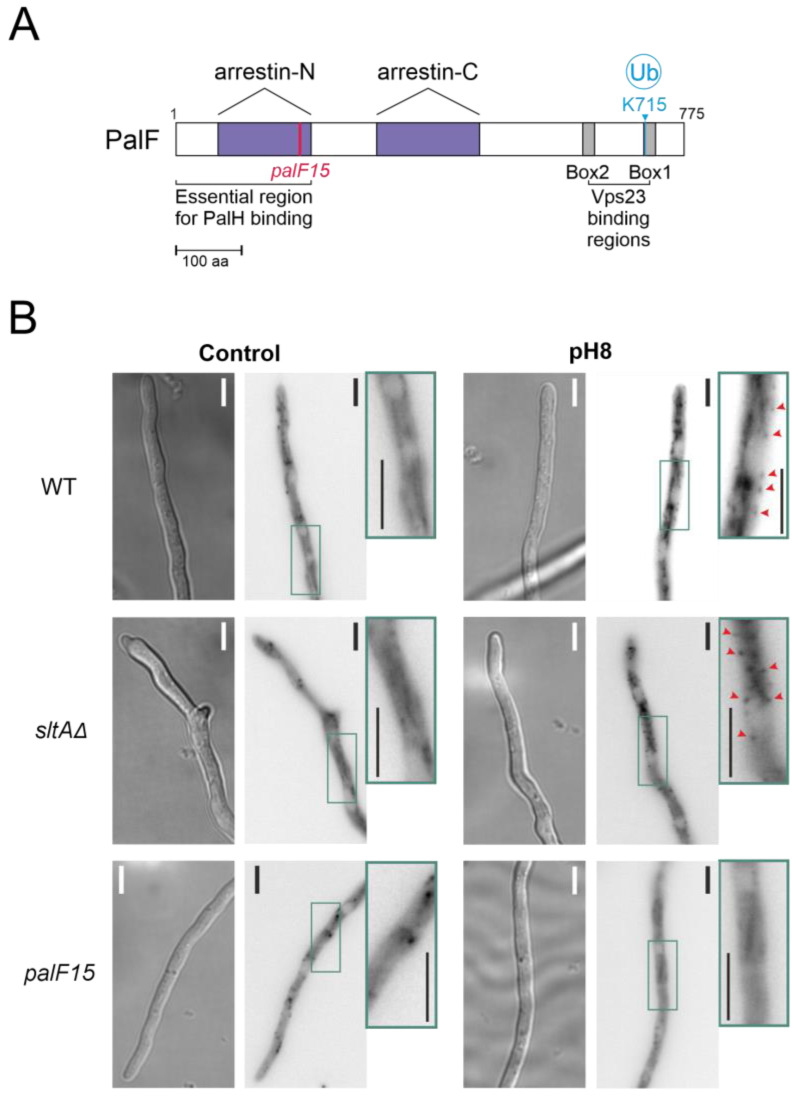
Recruitment of the Vps23 protein by post-translationally modified PalF protein at alkaline pH. (**A**) Schematic representation of PalF protein showing arrestin domains N and C (purple), the N-terminal PalH-binding region and the possible Vps23 binding regions (box 1 and box 2 in grey). Locations of the *palF15* mutation (pink) and the detected ubiquitination site at alkaline pH (light blue). (**B**) Fluorescence microscopy visualization of GFP-tagged Vps23 protein in standard (Control) and alkaline conditions (pH8) in WT (MAD7669) and null-*sltA* (MAD7670) strains. The *palF15* mutant (MAD3369) was used as an additional control to show Vps23 mislocalization at pH 8. The three strains were grown for 15 h at 25 °C in WMM. For alkalinisation assays, the WMM was replaced by WMM containing 100 mM Na_2_HPO_4_. Arrowheads indicate static cortical Vps23 spots visualized in images that are a sum projection of a time-lapse stack (interval 1 s for 46 s) following the methodology previously described [22]. Bar scale represents 5 µm.

**Figure 6 cells-13-00651-f006:**
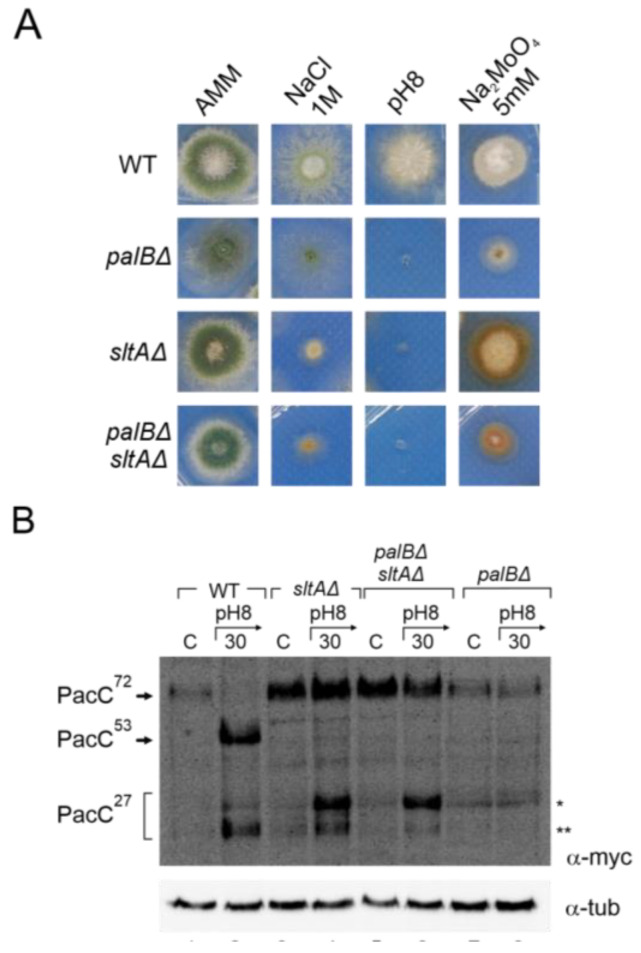
Effect of double null *palB* and *sltA* mutants in PacC proteolytic pattern and environmental stress phenotype. (**A**) Phenotype of wild-type strain (MAD7627), single mutants null *palB* and null *sltA* (MAD1775 and MAD7630), and of double mutant *palBΔ sltAΔ* (MAD8108) inoculated in alkaline and high concentration of sodium and molybdate conditions. Strains were grown at 37 °C for 48 h and colonial growth was imaged. (**B**) Immunodetecion of PacC proteolytic pattern in acid and alkaline-induced conditions in the strains indicated in panel (**A**). α-tubulin was used as loading control protein. * Indicates the band corresponding to very low mobility form of PacC fully proteolysed version, ** indicates the band corresponding to the standard PacC^27kDa^ proteolysed form, see the text for additional information.

**Figure 7 cells-13-00651-f007:**
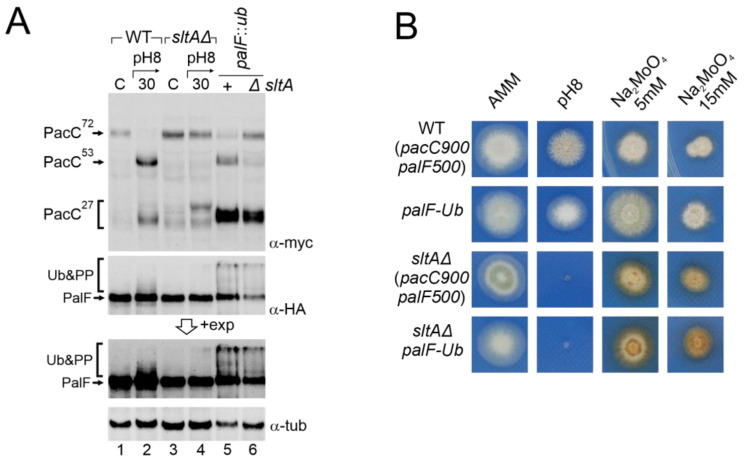
Effect of constitutive activation of the Pal pathway through PalF ubiquitination in a null *sltA* background. (**A**) Comparative patterns of PTMs visualized by the immunodetection of tagged forms of PacC and PalF in wild-type and *sltAΔ* background (MAD7627 and MAD7630), and in wild-type and *sltAΔ* strains (MAD4499 and MAD7634) carrying the *palF::ub* construct that expresses a constitutively ubiquitinated version of PalF. A longer exposure of PalF immunoblot is shown (+exp) to better show the presence or absence of ubiquitinated and phosphorylated forms (Ub&PP). α-tubulin was used as the loading control protein. (**B**) Phenotypic analysis of strains used in panel (**A**) tested under alkaline pH conditions and two sodium molybdate concentrations. Strains were grown at 37 °C for 48 h and colonial growth was imaged. Numbers indicate tracks cited in the text.

**Figure 8 cells-13-00651-f008:**
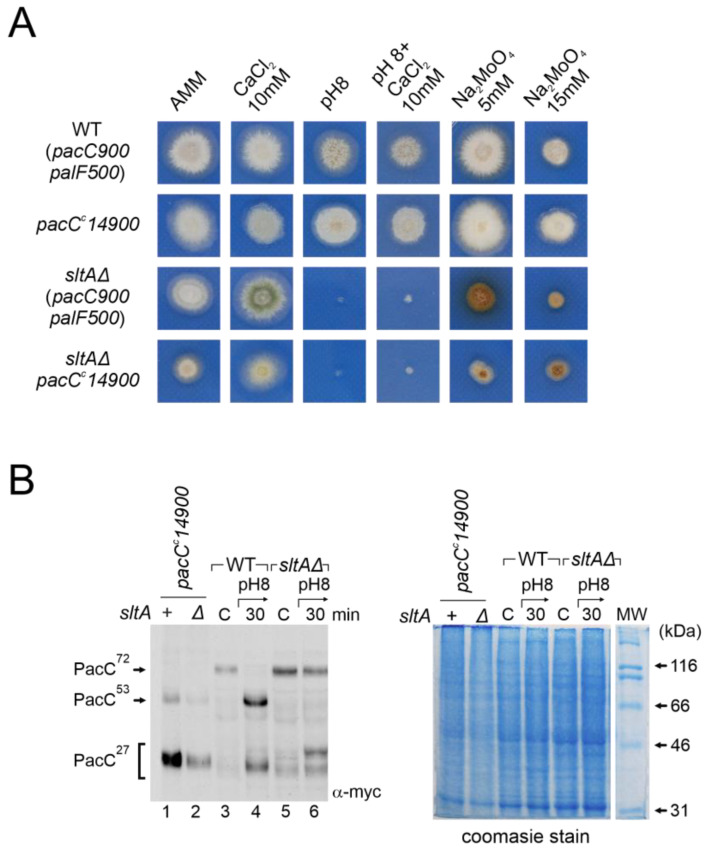
Analysis of the ambient pH-independent mutant *pacC^c^14900* in the null *sltA*. (**A**) Comparative colonial growth of *pacC^c^14900 sltAΔ* double mutant (MAD4296) with that shown by the wild-type (WT), *sltAΔ* and single *pacC^c^14900* mutant (MAD1445) strains on AMM containing different abiotic stresses. (**B**) Immunodetection of myc-PacC forms in the strains used in (**A**). (**B**, right) Coomassie staining blue was used for total protein loading control due to detection problems of α-tubulin in *pacC^c^14900* mutants. Numbers indicate tracks cited in the text.

**Figure 9 cells-13-00651-f009:**
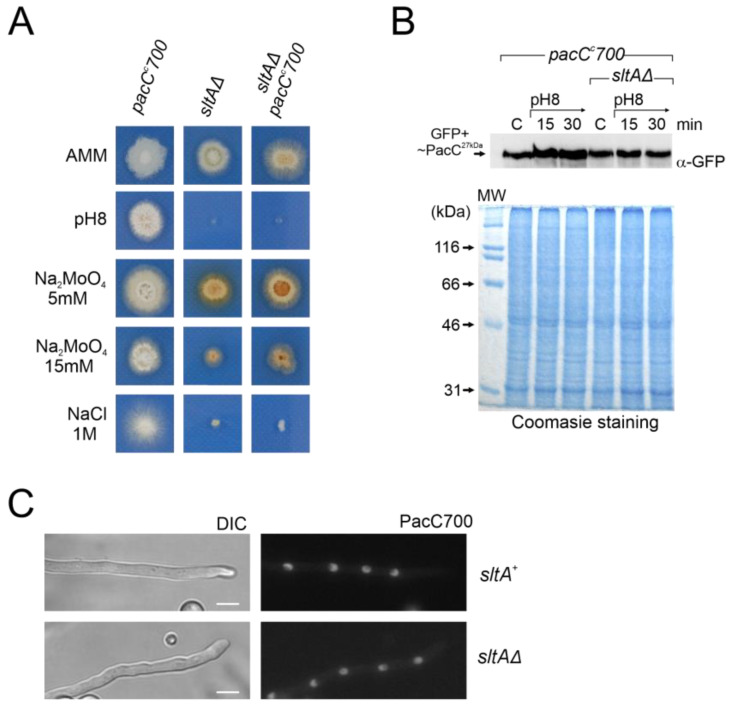
Effect of *pacC^c^700* allele in a null *sltA* background. (**A**) Phenotypes of single mutant *pacC^c^700* (GFP-PacC5-250) (MAD7632) and *sltAΔ* (MAD3816) strains and the *pacC^c^700 sltAΔ* double mutant (MAD7633). Strains were incubated at 37 °C for 48 h and colony growths were imaged. (**B**) Immunodetection of PacC700 in a wild-type (*sltA*^+^) and the null *sltA* background. As in Figure 7, Coomassie staining was used as the protein loading control. (**C**) Fluorescence microscopy of WT and null *sltA* strains expressing GFP-PacC5-250 (PacC700) in WMM showing the accumulation of fluorescence in nuclei. DIC indicates Nomarsky optics for cell visualization. Scale bar represents 5 μm.

**Figure 10 cells-13-00651-f010:**
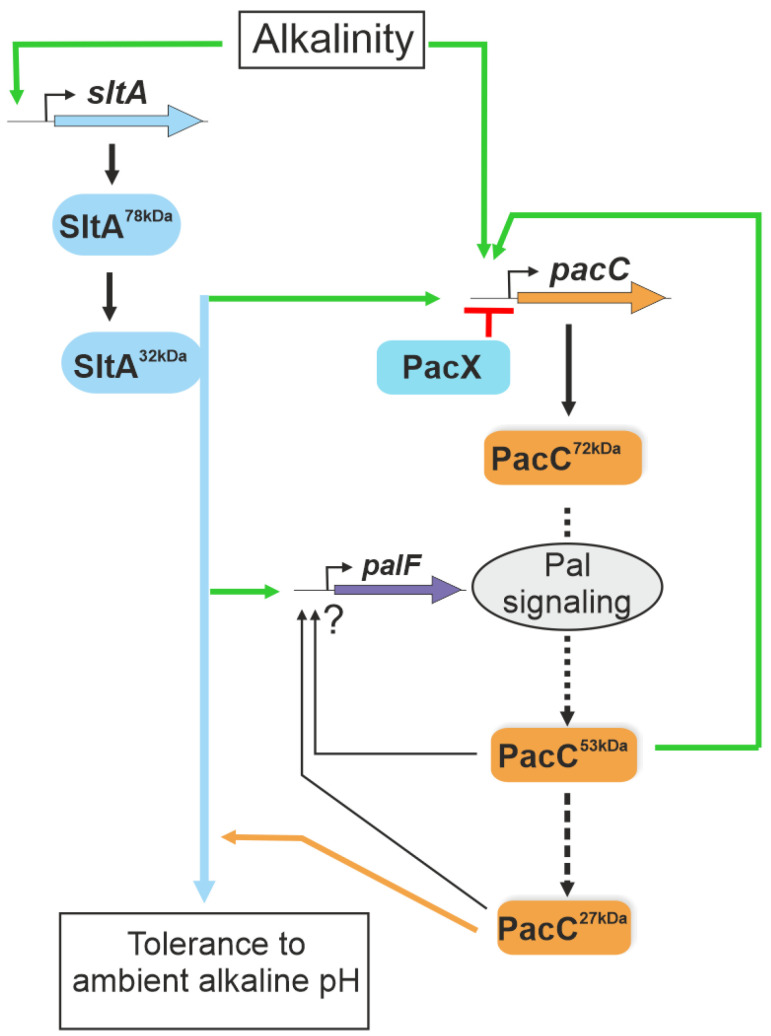
A model of the interconnection between Slt and Pal/PacC systems in the transcriptional response to ambient alkaline pH. The scheme integrates data from Bussink and collaborators [29] and the results obtained in this study. SltA is proposed to positively act (green arrows), regulating *pacC* expression and the correct processing of the PacC protein via the activation of the Pal signalling pathway at alkaline pH. SltA is epistatic to PacC in regulating the tolerance to ambient alkaline pH (blue and orange lines). Green lines indicate positive actions by the regulators and in red the negative activity of PacX. The ? symbol indicates undefined regulatory activities by PacC proteolysed forms.

## Data Availability

Data are contained within the manuscript and Appendix A.

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
