# Peer review of "Interconnections between the Cation/Alkaline pH-Responsive Slt and the Ambient pH Response of PacC/Pal Pathways in Aspergillus nidulans"

_cells, 2024, doi:10.3390/cells13070651_

Round 1
Reviewer 1 Report
Comments and Suggestions for Authors
In the present manuscript the authors describe the interplay between the Stl and PacC signaling pathways regulating tolerance to ambient alkalinity. Overall, the text is well written and clear, the results are very interesting, showing novelties in the role of these proteins in ambient pH adaptation in the model fungi Aspergillus nidulans. Some modifications, mainly statistical analysis for the RT-qPCR is highly recommended to improve results and conclusions.
In the abstract (lines 23-25), the authors conclude that “StlA function is hierarchically superior to PacC on regulating gene expression for tolerance to ambient alkalinity”. Although PacC is a transcription factor and indeed regulates gene expression, in the present work, its regulon was not evaluated. Thus, to avoid misleading the readers to wrong conclusions, I suggest changing this statement to “StlA function is hierarchically superior to PacC on regulating tolerance to ambient alkalinity in A. nidulans”.
Lines 124 and 126, change to “cleistothecia”.
I recommend the authors to perform statistical analysis for all the RT-qPCR assays and show them in the graphs to highlight the differences that are discussed in the text. It is not clear some of the conclusions about gene expression, due to the lack of statistical analysis and use of proper reference samples. Overall, it seems that the control, acidic condition, of each strain was used as the reference. However, when analyzing the expression of a given gene (pacC and palF in Figure 1), in the WT vs deletion strains, the use of the WT as reference would make the results clearer and really show the difference between the strains and a better comprehension of this regulation, given that the expression in acidic condition may also differ between them. In lines 237-239 the authors mention the upregulation of palF in the null stl strain, but it is does not seems to be statistical different only by looking at the graphs (statistical analysis is required). Line 340 and 346 – “the expression is slightly higher”, and “pacX expression increased”, without statistics it is not possible to state that. In Figure S3A, the profile is different, but standard deviation is also high. Then, with the statistics, please revise the statements regarding gene expression in the text.
In Figure 2, the PalF PTMs are not clear, as they are in Figure 4D.
Author Response
In the present manuscript the authors describe the interplay between the Stl and PacC signaling pathways regulating tolerance to ambient alkalinity. Overall, the text is well written and clear, the results are very interesting, showing novelties in the role of these proteins in ambient pH adaptation in the model fungi Aspergillus nidulans. Some modifications, mainly statistical analysis for the RT-qPCR is highly recommended to improve results and conclusions.
In the abstract (lines 23-25), the authors conclude that “StlA function is hierarchically superior to PacC on regulating gene expression for tolerance to ambient alkalinity”. Although PacC is a transcription factor and indeed regulates gene expression, in the present work, its regulon was not evaluated. Thus, to avoid misleading the readers to wrong conclusions, I suggest changing this statement to “StlA function is hierarchically superior to PacC on regulating tolerance to ambient alkalinity in A. nidulans”.
We appreciate the reviewer’s positive feedback on our manuscript. We have implemented her/his suggestions for modifying the abstract. As both reviewers expressed concern about the same sentence in the abstract, we have rewritten this final statement to accommodate their suggestions. Furthermore, we have made significant revisions to the English style based on the other reviewer’s suggestions.
Lines 124 and 126, change to “cleistothecia”.
Corrected as indicated
I recommend the authors to perform statistical analysis for all the RT-qPCR assays and show them in the graphs to highlight the differences that are discussed in the text. It is not clear some of the conclusions about gene expression, due to the lack of statistical analysis and use of proper reference samples. Overall, it seems that the control, acidic condition, of each strain was used as the reference. However, when analyzing the expression of a given gene (pacC and palF in Figure 1), in the WT vs deletion strains, the use of the WT as reference would make the results clearer and really show the difference between the strains and a better comprehension of this regulation, given that the expression in acidic condition may also differ between them.
Following the reviewer’s suggestion, we have performed the indicated statistical analyses. We included a brief description of these analyses in the Materials and Methods section. Asterisks indicating the range of p-values and the level of significance have been added to each figure legend and in the text where appropriate.
Our main aim in using the data shown in Figure 1 was to demonstrate the variation in expression profiles between the WT and null sltA background over the times tested. qPCRs were limited to a specific number of samples due to the complexity of the experiment, which prevented the inclusion of all samples in a single plate with the appropriate replicates and controls. However, in Figures 4A and 4B, we show the differences between WT and null sltA backgrounds in the expression levels of PacC and PalF at acidic pH, among other comparisons. The text has been revised to describe how we compare the effect of lack of SltA function on the expression profiles of the pacC and palF genes in Figure 1.
In lines 237-239 the authors mention the upregulation of palF in the null stl strain, but it is does not seems to be statistical different only by looking at the graphs (statistical analysis is required).
We have included the statistical analysis and modulated our comments on this profile. Although expression levels of palF seemed to elevate along 15 and 30 min after exposure to alkaline pH, these values were not statistically significant.
Line 340 and 346 – “the expression is slightly higher”, and “pacX expression increased”, without statistics it is not possible to state that.
The text has been modified in accordance with the statistical analyses.
In Figure S3A, the profile is different, but standard deviation is also high. Then, with the statistics, please revise the statements regarding gene expression in the text.
This section of the manuscript has undergone thorough revision. The qPCR measurements revealed significant variations in pacX expression levels. We have revised the data and added a new paragraph to cover the experiments presented in Figure S3 and Figure 3. Consequently, panel A from Figure S3 has been incorporated into Figure 3, and the previous panel B in Figure 3 is now panel C.
In Figure 2, the PalF PTMs are not clear, as they are in Figure 4D.
We agree with the reviewer that the PalF modified forms are more clearly visible in the pacX20 mutant background, most likely due to the higher levels of PalF protein. We attempted to modify the overexposed WB image to better display the smear corresponding to the PalF modified forms in the wild-type background, while avoiding excessive image processing without clear improvement. We have modulated the text describing this observation to focus the reader on the work done with the constitutively ubiquitinated version of PalF.
Reviewer 2 Report
Comments and Suggestions for Authors
Introduction does a fairly good job of laying out the understanding of regulation of fungal stresses, with a focus on alkaline pH stress on A. nidulans. One point that could help with improving comprehension would be a model figure, either as part of the introduction, or to describe the final conclusions for the positioning of SltA and PacC function.
The initial data involve investigating SltA involvement in the transcriptional activation of genes in response to external alkalinization. Loss of SltA reduced PacC expression in response to pH increase, and modified PalF expression. Further experiments were directed at the relationship between SltA and PacC and PalF activity. In strains that lacked SltA, the normal proteolytic processing of PacC in response to an alkali signal was disrupted. Disappearance of the PacC72 band is rapid in the WT and very slow in the sltA mutant; appearance of the PacC52 band rapid in WT and essentially non-existent in the sltA mutant, and the distribution of bands designated PacC27 is also different between the WT and the sltA mutant. Even mild sltA mutants affect processing, suggesting SltA influences both expression and proteolytic processing of PacC.
Next the authors investigate the impact on PalF, which is typically targeted for post translational modifications during activation of the alkaline response, and is required for the proteolytic processing of PacC. No post translational higher molecular weight “smearing” of the protein band is detected in the sltA mutant background, consistent with PalF failing to undergo post-translational modifications in the sltA mutant. These results are convincing, and the issues with PalF could potentially explain the failure in proteolytic processing of PacC.
Further experiments investigate a role for PacX in the circuitry. In WT cells loss of PacX enhances PacC expression, but it does not rescue loss of SltA function. These assays are both functional, and direct transcriptional studies; in both assays it is clear that the sltA mutant phenotypes are not suppressed by deletion of PacX. PacX expression itself is influenced by deletion of SltA
A major concern for this paper is the overall novelty of the results. In many cases the results are repeating findings from previous papers, often from the current authors, and are just improving the resolution of the data but not providing new insights. This manuscript should emphasize what is truly new, and clearly acknowledge where the results are repeating prior analyses at a better resolution.
Comments on the Quality of English Language
Overall paper can benefit somewhat from editing for English. Some phrases are not idiomatic, and this can lead to confusion in interpretation.
In the abstract, “hierarchically superior” means, I think, “acts prior to” – the latter statement would be clearer to an English speaker. While “superior” can mean prior or above in a pathway or hierarchy, it is usually not linked with “acting”, but just designating an order or position, “acting superior” gets used elsewhere to mean acting before or acting prior. The following are just a few examples of where English style could be improved
17 of PacX factor. of the PacX factor.
74 As PacC, SltA displays three forms As also seen with PacC…
276 lost-of-function mutants loss of function
318 truncation of SltA after zinc finger truncation of SltA after the zinc finger
609 not only affects sltB expressing in a great manner but also a significant number of genes [26].
Not only affects high level sltB expression…
612 Our results evidenced a need of SltA function to maintain pacX transcription and the up-regulatory effect of lack
Our results showed a need…
Author Response
Introduction does a fairly good job of laying out the understanding of regulation of fungal stresses, with a focus on alkaline pH stress on A. nidulans. One point that could help with improving comprehension would be a model figure, either as part of the introduction, or to describe the final conclusions for the positioning of SltA and PacC function.
The initial data involve investigating SltA involvement in the transcriptional activation of genes in response to external alkalinization. Loss of SltA reduced PacC expression in response to pH increase, and modified PalF expression. Further experiments were directed at the relationship between SltA and PacC and PalF activity. In strains that lacked SltA, the normal proteolytic processing of PacC in response to an alkali signal was disrupted. Disappearance of the PacC72 band is rapid in the WT and very slow in the sltA mutant; appearance of the PacC52 band rapid in WT and essentially non-existent in the sltA mutant, and the distribution of bands designated PacC27 is also different between the WT and the sltA mutant. Even mild sltA mutants affect processing, suggesting SltA influences both expression and proteolytic processing of PacC.
Next the authors investigate the impact on PalF, which is typically targeted for post translational modifications during activation of the alkaline response, and is required for the proteolytic processing of PacC. No post translational higher molecular weight “smearing” of the protein band is detected in the sltA mutant background, consistent with PalF failing to undergo post-translational modifications in the sltA mutant. These results are convincing, and the issues with PalF could potentially explain the failure in proteolytic processing of PacC.
Further experiments investigate a role for PacX in the circuitry. In WT cells loss of PacX enhances PacC expression, but it does not rescue loss of SltA function. These assays are both functional, and direct transcriptional studies; in both assays it is clear that the sltA mutant phenotypes are not suppressed by deletion of PacX. PacX expression itself is influenced by deletion of SltA
A major concern for this paper is the overall novelty of the results. In many cases the results are repeating findings from previous papers, often from the current authors, and are just improving the resolution of the data but not providing new insights. This manuscript should emphasize what is truly new, and clearly acknowledge where the results are repeating prior analyses at a better resolution.
We appreciate the reviewer's positive feedback on our manuscript and we have implemented all suggested improvements. Specifically, we have contextualised our new data using state-of-the-art techniques with previous published results. We have also included a model (Figure 10) summarising our findings as suggested by the reviewer.
Comments on the Quality of English Language
Overall paper can benefit somewhat from editing for English. Some phrases are not idiomatic, and this can lead to confusion in interpretation.
In the abstract, “hierarchically superior” means, I think, “acts prior to” – the latter statement would be clearer to an English speaker. While “superior” can mean prior or above in a pathway or hierarchy, it is usually not linked with “acting”, but just designating an order or position, “acting superior” gets used elsewhere to mean acting before or acting prior.
The English style has been corrected as suggested. The changes made are indicated in the text following the Editor’s instructions. As both reviewers expressed concern about a sentence in the abstract, we have revised it to address their suggestions.
Additionally, all of the changes listed below have been implemented.
The following are just a few examples of where English style could be improved
17 of PacX factor. of the PacX factor.
74 As PacC, SltA displays three forms As also seen with PacC…
276 lost-of-function mutants loss of function
318 truncation of SltA after zinc finger truncation of SltA after the zinc finger
609 not only affects sltB expressing in a great manner but also a significant number of genes [26].
Not only affects high level sltB expression…
612 Our results evidenced a need of SltA function to maintain pacX transcription and the up-regulatory effect of lack
Our results showed a need…